# Better Call Graphs: A New Dataset of Function Call Graphs for Malware Classification

## Abstract

Malware classification by using function call graphs (FCG) is an important task in cybersecurity. One big challenge in this direction is the lack of representative, large, and unique FCG datasets. Existing datasets typically contain obsolete Android application packages (APKs), largely consist of small graphs, and include many duplicate FCGs due to repackaging. This results in misleading graph classification performance. In this paper, we propose a new comprehensive dataset, Better Call Graphs (BCG), that contains large and unique FCGs from recent APKs, along with graph-level APK features, with benign and malware samples from different types and families. We establish the necessity of BCG through the evaluation of several baseline approaches on existing datasets. BCG is available at `https://iclr.me/`.

## 1 Introduction

Malware detection is a key task in the field of cybersecurity. In most malware samples, minor changes in the source code of the original malware can lead to substantially different compiled code (e.g., through instruction reordering, branch inversion, and register allocation) (Bayer et al., 2006). This is often exploited to bypass signature-based detection, a common method of malware detection (Scott, 2017). However, these minor source code modifications have little impact on the executable's control flow, which can be depicted using a function call graph (FCG) where functions are the nodes and the call relations are the edges. Hence FCG-based malware detection has been an important field of study, particularly within the realm of Android (Ye et al., 2017; Freitas et al., 2020).

One challenge in FCG-based malware detection is that achieving accurate and robust classification models has been hampered by the limited availability of modern and representative large-scale datasets. Existing datasets typically contain old Android application packages (APKs) along with their corresponding FCGs. Considering the dynamic landscape of Android ecosystem, obsolete APKs offer no benefit at all as they are developed on earlier version of an Android. In addition, the complexity of both benign and malicious applications has drastically changed in recent years. Furthermore, most existing datasets contain many duplicate APKs, packaged with trivial differences, hence have a different name but the same FCG structure.

In this paper, we introduce a new and comprehensive dataset named Better Call Graphs (BCG), comprising extensive and distinct function call graphs (FCGs) extracted from recent APKs. The dataset includes benign samples as well as malware samples spanning various types and families. We establish the necessity of BCG through the evaluation of several baseline approaches on existing datasets. We show that existing datasets often yield misleading scores when state-of-the-art classifiers are applied. Our dataset also contains graph-level APK features, capturing both the structural and behavioral characteristics of malware. BCG is publicly available at `https://iclr.me/`. The dataset is released under a CC-BY license, enabling free sharing and adaptation for research or development purpose.

In the rest of the paper, we first give a background and summarize related works on FCG-based Android malware detection and Android-based FCG datasets in Section 2. Then we provide a detailed description of how the BCG is collected and filtered in Section 3. Next, we explain the properties of BCG in detail (Section 4) and perform graph classification experiments on our dataset, as well as other established datasets, using state-of-the-art graph classification methods (Section 5) — we

perform malware type as well as family classification. Finally, we summarize our work and discuss the limitations in Section 6.

## 2   RELATED WORK

Literature is rich with studies on Android malware detection, by using various types of networks such as Function Call Graphs (FCGs), Control Flow Graphs, and Network Flow Graphs. These studies rely on existing malware datasets to evaluate their methods' effectiveness. Here we delve into prior research on FCG-based detection for Android apps, followed by a discussion of commonly used malware datasets.

### 2.1   FCG-BASED ANDROID MALWARE DETECTION

Android apps, packaged in APK files, bundle all their components — code, resources, and a manifest file. Extracting various features from the code allows researchers to analyze how the app works and identify potential security risks. FCGs, in particular, have proven valuable for malware classification by revealing how the app's functions interact, potentially exposing malicious behavior. Researchers have actively explored diverse methods that leverage FCGs to analyze Android apps for security purposes. These methods typically involve constructing FCGs and enriching them with node features, which can be either basic properties or more complex embeddings learned through graph neural networks. The enriched FCGs are then used for identifying and classifying malware. Classifications are typically performed to identify both the malware type and the malware family. Malware type refers to the broad category of malicious behavior exhibited by a malware program. Common types of malware include Virus, Worm, Trojan Horse, and Ransomeware. Malware family refers to a group of malware programs that share similar characteristics, codebase, or functionality. Malware families are often named by antivirus companies based on their unique features. For example, within the "Trojan Horse" type, there might be the "Emotet" family known for email spam and credential theft.

MAMADroid and APIGraph utilized API semantics features to capture the semantic similarities between malware variants and analyze information flow for malware detection (Onwuzurike et al., 2019; Zhang et al., 2020). In contrast, Yuan et al. (2020) focused on byte-level classification by converting malware binaries into Markov images and applying deep learning for detection. Meanwhile, Fan et al. (2018) proposed a family-level classification approach by leveraging frequent subgraphs within FCGs. Zhu et al. (2018) constructed enriched FCGs from Smali code, incorporating function types (system or third-party API) and permission requirements for each node, and used Graph Convolutional Networks (GCNs) to train malware classifiers on these graphs. Similarly, Feng et al. (2020) focused on extracting features directly from the disassembled code sections in CGdroid. This approach first uses hand-crafted features like the number of string constants and instructions for each node, and then utilizes a GNN to learn graph embeddings and an MLP for final classification. Vinayaka & Jaidhar (2021) further extended this concept by incorporating FCG's graph structural attributes (e.g., node degree) and non-graph features extracted from the disassembled functions, such as method attributes and opcode summaries. Yumlembam et al. (2022) took a more general approach, modeling apps as local graphs where nodes denote APIs and co-occurring APIs in the same code block as edges. The authors explored features like centrality measures, permissions, and intents from the manifest file. Lo et al. (2022) leveraged PageRank (Page et al., 1999), in/out degree, and betweenness centrality values as node attributes. DeepCatra utilized call traces, opcode features, and TF-IDF for critical API identification (Wu et al., 2023).

Moving beyond basic features, some approaches explored learning node embeddings using GNNs and NLP techniques (Catal et al., 2021; Gunduz, 2022). Errica et al. (2021) leveraged Contextual Graph Markov Models to learn embeddings based on call graph structure and out-degree features, followed by classification with a neural network. Cai et al. (2021) and Xu et al. (2021) have explored leveraging word embedding techniques to analyze Android apps. Xu et al. (2021) used the Skip-gram algorithm to transform Android opcodes into vectors for analysis.

These approaches highlight the active research in utilizing FCGs for Android app security analysis. **However, a crucial limitation of most existing studies is the reliance on datasets that contain obsolete and/or duplicate APKs.** This inflates the reported performance of classification approaches. Our work addresses this limitation by introducing a new malware classification dataset, BCG, that

consist of new and unique FCGs. BCG will pave the way for true evaluation the effectiveness of the aforementioned FCG-based methods and open doors for new research directions in the field of Android app security analysis.

Table 1: Comparison of previous Android-based FCG datasets and BCG.

| Dataset | # APKs | Collection Period | # types | Family information |
|---|---|---|---|---|
| Drebin | 5560 | 2010-2012 | N/A | Yes (179, no benign) |
| AndroZoo | 24M | Dynamic | N/A | No |
| CICAndMal2017 | 10854 | 2015-2017 | 5 | No |
| CICMalDroid | 17341 | 2018 | 5 | No |
| MalNet | 1.2M | 2006-2021 | 47 | Yes (696) |
| MalNet-Tiny | 5000 | 2006-2021 | 5 | Yes (5) |
| **BCG (our work)** | 9938 | 2017-2023 | 29 | Yes (118) |

## 2.2 ANDROID-BASED FCG DATASETS

While numerous graph classification datasets exist for various fields like bioinformatics and social networks, options for cybersecurity, specifically malware detection using graph analysis, are scarce. Most existing datasets in this domain are closed-source. Fortunately, a few publicly available options like CICAndMal2017, CICMalDroid, AndroZoo, Drebin, and MalNet provide valuable resources for researchers analyzing Android malware through graph structures.

Given the dynamic nature of the Android ecosystem and the interest of malicious entities in releasing APKs, several FCG datasets have been collected and curated to study malware characteristics. The Drebin dataset is one of the first in this direction, offering 5,560 malware apps (179 families) collected between 2010-2012 by MobileSandbox (Arp et al., 2014), but it lacks benign samples. Drebin provides summaries of each malicious APK using 10 features like permissions and intents. While the Drebin dataset is valuable for multi-class classification (identifying the top-k most frequent malware families), binary classification (malware vs. benign) requires collecting benign samples from other sources. The AndroZoo dataset contains over 24 million APKs, mostly benign and with some malware verified through VirusShare (Allix et al., 2016). AndroZoo is constantly updated and provides 10 features per APK which includes sha256, sha1, md5, apk_size, dex_size, dex_date, pkg_name, vercode, vt_detection, and vt_scan_date. It primarily consists of benign APKs, and the fraction of malware APKs is 16.67% (4 million APKs). The Canadian Institute for Cybersecurity (CIC) released two Android app datasets for malware analysis: CICAndMal2017 (Lashkari et al., 2018) and CICMalDroid (Mahdavifar et al., 2020). The former one has 10,854 APKs collected between 2015-2017 and categorized by malware type (benign, adware, ransomware, SMS, and riskware) and includes network traffic features. The latter one boasts a larger collection, 17,341 APKs from 2018, with similar malware classifications. It provides not only static features (permissions, intents) but also dynamic behavior data (system calls) and network traffic (pcap format). MalNet is a more recent dataset of 1.2 million FCGs extracted from AndroZoo APKs collected between 2006-2021 (Freitas et al., 2020). The dataset is categorized into 47 types and 696 families. It also offers a smaller version with 5000 FCGs, MalNet-Tiny, for efficient experimentation. Another related dataset, MalRadar, includes 4,534 APKs collected from 2014 to 2021 and provides information only at the family level (Wang et al., 2022).

While these datasets are valuable, they suffer from two key issues:

1. Most FCGs are obtained from **obsolete APKs**, latest being only from 2021 (most of which are also benign). The complexity of both benign and malicious applications has drastically changed in recent years. For example, in the past it was simple enough to classify malware by determining if phone SIM card details were sent over the network. However, now benign applications do this for two-factor authentication and even for unique tracking of individuals for leaderboards in games.
2. The current datasets often contain many **duplicate APKs**, packaged with trivial differences, which potentially inflates the classification performance (details in Section 4.3).

In this work, we address these limitations by constructing BCG, a new dataset with unique and recent APKs filtered with respect to various criteria, to enable a more robust testbed for FCG-

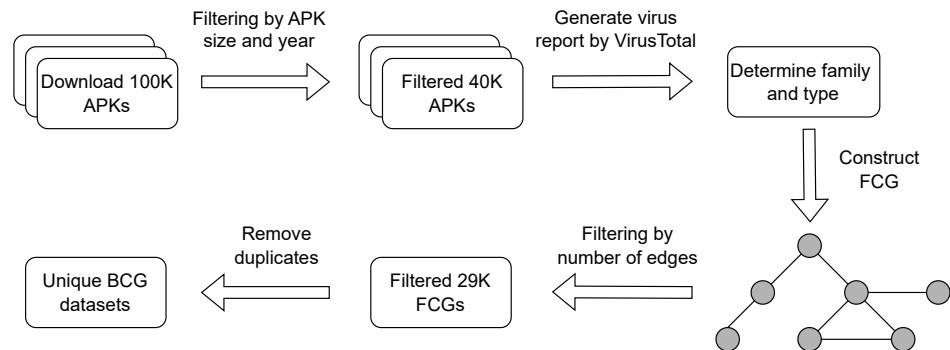

Figure 1: Construction process of BCG.

based malware classification. Those criteria include the minimum APK size to exclude very small applications, minimum number of edges in the FCG to focus on complex functionalities, and multi-engine validation via VirusTotal (vir, 2024b) to guarantee high-confidence malware labels. A detailed comparison between our work and existing datasets is presented in Table 1.

## 3 COLLECTING AND FILTERING FCGS

To ensure a robust and relevant BCG dataset, we constructed it by comprehensively analyzing both APK files and their corresponding graph properties. This involved filtering and refining APKs based on various quality and relevance criteria, overcoming limitations of existing datasets. There are some high level observations that guided our approach: (1) old APKs are simplistic in their structure and capabilities, (2) repackaging is very common, (3) certain virus families are over represented in datasets (often related to point 2), and (4) small (based on bytecode size and not auxiliary files) APKs are often uninteresting from a detection standpoint. A detailed flowchart of the BCG construction process is provided in Figure 1. Here we summarize each step in this process:

**1. Downloading APKs:** The foundation of our BCG dataset is built upon acquiring data sources that exhibit specific properties, including larger and more recent APKs (details of these properties are provided in Section 4). To achieve this, we secured permission from two prominent repositories: AndroZoo (Allix et al., 2016) and VirusShare (vir, 2024a). We selected AndroZoo and VirusShare as representative sources due to their extensive collections and widespread use in the research community. These are the defacto sources for the mobile, programming languages, and Android security communities. AndroZoo, for example, includes apps from the Google Play Store as well as other app sources such as Appchina and Anzhi (Chinese App Store), encompassing a total of 24,751,611 APKs—significantly more than the number of apps currently available on the Google Play Store. This extensive repository covers all apps from these sources without any exclusions. We downloaded over 100,000 APKs from the AndroZoo and VirusShare repository. Due to a limitation within AndroZoo that restricts concurrent downloads to 40 files, this process took around 72 hours to complete. Additionally, storing this vast amount of data required 2 TBs of storage space.

**2. Filtering APKs by year and size:** To ensure our dataset has recent and large files, we followed a two-step filtering process on the 100,000 downloaded APKs. Firstly, we extracted the DEX (Dalvik Executable) year from each APK. DEX year refers to the approximate year the application was compiled and the information is embedded within the APK file. By removing APKs with a DEX year before 2017, we ensured our dataset primarily reflects more recent applications. Secondly, to ensure the APKs contained sufficient information for analysis, we removed any APKs with a size less than 4MB. We analyzed the distribution of APK size alongside the number of nodes and edges in their FCGs. This analysis revealed that APKs exceeding 4MB typically contained enough nodes and edges for meaningful FCG-based analysis. Consequently, we filtered out APKs with a size less than 4MB. These filtering process resulted in a dataset of around 40,000 APKs.

**3. Determining family and type:** To create the types and families in BCG, we utilized a multi-step process with VirusTotal (vir, 2024b). First, we downloaded detailed reports for each APK using the

Table 2: Descriptive statistics for each FCG type in BCG.

| Type | #graphs | #fams | # nodes | | | | # edges | | | |
|---|---|---|---|---|---|---|---|---|---|---|
| | | | min | mean | max | std | min | mean | max | std |
| benign | 5880 | 1 | 78 | 30.1K | 50.4K | 12.1K | 109 | 62.6K | 100.0K | 26.3K |
| trojan | 1525 | 83 | 90 | 18.8K | 47.5K | 13.4K | 101 | 41.8K | 99.9K | 30.5K |
| adware | 803 | 62 | 92 | 23.3K | 47.7K | 13.2K | 101 | 51.5K | 99.9K | 30.1K |
| smsreg | 279 | 22 | 96 | 21.8K | 41.9K | 9.9K | 106 | 49.1K | 99.3K | 22.7K |
| adware++trojan | 250 | 40 | 92 | 20.0K | 47.3K | 14.4K | 101 | 44.5K | 99.9K | 32.6K |
| riskware | 157 | 19 | 87 | 15.7K | 46.0K | 14.1K | 103 | 34.9K | 99.4K | 30.9K |
| smsreg++trojan | 121 | 20 | 81 | 21.7K | 45.3K | 11.3K | 103 | 49.9K | 98.5K | 26.0K |
| riskware++trojan | 119 | 22 | 97 | 13.0K | 44.7K | 12.4K | 103 | 28.3K | 97.0K | 27.6K |
| risktool++trojan | 114 | 18 | 97 | 12.2K | 47.7K | 12.8K | 101 | 24.4K | 96.7K | 26.8K |
| addisplay | 82 | 9 | 235 | 26.8K | 44.8K | 8.8K | 351 | 58.5K | 99.4K | 22.6K |
| dropper++trojan | 74 | 13 | 93 | 14.8K | 46.1K | 13.1K | 112 | 31.6K | 91.5K | 27.7K |
| risktool | 67 | 17 | 90 | 15.7K | 42.0K | 13.5K | 103 | 33.2K | 95.8K | 29.5K |
| spy++trojan | 57 | 16 | 101 | 16.2K | 46.5K | 15.0K | 110 | 35.0K | 99.8K | 33.0K |
| banker++trojan | 54 | 3 | 77 | 7.8K | 44.5K | 11.2K | 118 | 17.5K | 99.7K | 25.1K |
| adware++riskware | 47 | 15 | 97 | 13.8K | 41.7K | 14.1K | 103 | 31.2K | 96.4K | 32.3K |
| risktool++riskware | 47 | 10 | 90 | 9.8K | 44.3K | 12.5K | 103 | 20.0K | 95.8K | 26.4K |
| spr++trojan | 35 | 12 | 101 | 16.8K | 44.3K | 13.0K | 110 | 39.8K | 98.0K | 32.6K |
| riskware++smsreg | 30 | 9 | 101 | 17.2K | 45.6K | 13.3K | 110 | 38.6K | 92.1K | 29.8K |
| rog | 30 | 3 | 97 | 5.5K | 43.2K | 11.1K | 103 | 12.2K | 92.3K | 23.2K |
| smsreg++spr | 30 | 4 | 589 | 26.9K | 46.2K | 10.7K | 762 | 62.7K | 96.3K | 25.5K |
| downloader++trojan | 29 | 10 | 97 | 17.3K | 44.9K | 16.2K | 103 | 38.6K | 92.1K | 35.6K |
| spy | 27 | 11 | 548 | 18.9K | 43.1K | 14.4K | 1561 | 42.0K | 98.0K | 33.5K |
| clicker++trojan | 26 | 4 | 119 | 22.7K | 38.8K | 13.2K | 124 | 54.9K | 93.4K | 33.3K |
| fakeapp++trojan | 13 | 3 | 119 | 16.4K | 45.2K | 15.6K | 124 | 32.3K | 95.8K | 34.0K |
| risktool++spr | 13 | 6 | 98 | 16.5K | 27.8K | 10.5K | 103 | 35.0K | 65.7K | 24.1K |
| fakeapp | 8 | 2 | 119 | 24.0K | 45.2K | 15.7K | 124 | 47.9K | 95.8K | 35.1K |
| adware++risktool | 7 | 3 | 2782 | 23.5K | 44.7K | 15.5K | 9409 | 53.7K | 99.0K | 33.8K |
| backdoor | 7 | 3 | 109 | 9.8K | 35.3K | 16.5K | 112 | 21.2K | 82.9K | 36.4K |
| backdoor++trojan | 7 | 2 | 98 | 0.1K | 0.1K | 0.0K | 103 | 0.1K | 0.2K | 0.0K |

VirusTotal API, leveraging over 70 antivirus engines. We then extracted virus categories and families from these reports. To obtain a single, consistent label, we converted the VirusTotal report to one compatible with the AVClass package, which assigned a single label from the reported information. To ensure reliability, we only consider the APKs flagged by multiple antivirus engines in VirusTotal.

**4. Constructing FCGs:** Androguard has proven effective in previous datasets for constructing FCGs (Desnos & Gueguen, 2011). It conducts static analysis of the DEX file within each APK, identifying method names and their interactions. Methods are represented as nodes, while calls between methods are depicted as directed edges. We leverage Androguard to produce FCGs for each APK in our dataset. To facilitate further analysis, we also hash the method names into unique identifiers, enabling the creation of an integer edge list. Our datasets are published in two formats: one containing the original method names as nodes, and another using hashed IDs.

**5. Filtering APKs by number of edges:** While the MalNet dataset has a large average graph size, it contains many very small graphs (less than 100 nodes/edges). To exclude trivially small graphs (less than 100 nodes/edges, as observed in MalNet), we further filtered our datasets to only include APKs with a minimum of 100 edges in their FCGs, which is large enough to contain complex graph structure. This filtering process resulted in a collection of around 29K graphs from 40K graphs.

**6. Removing duplicates:** Existing datasets, such as MalNet and CICMaldroid, often contain duplicate APKs with identical functionalities (reflected in their FCG properties like number of nodes, edges, etc.) but disguised by different names. This can be due to repackaging (malware with minor changes) or the same APK appearing in multiple app stores. To address this, we identified and removed duplicate FCGs based on six key graph properties of the FCGs: number of nodes, number of edges, average degree, in-degree centrality, size of largest connected component, and size of largest weakly connected component. For further confirmation, we verified that these duplicates also shared the same malware type and family information. Our methodology results in no false positives but there may well remain false negatives, i.e., undetected duplicates or FCGs that differ by one or few

nodes/edges, which is an interesting future work. The APKs with identical properties are considered duplicates and removed, resulting in a non-redundant final BCG dataset.

# 4 PROPERTIES OF BCG

While existing datasets like MalNet (Freitas et al., 2020) offer valuable properties for evaluating Android malware with FCGs, they often contain duplicate APKs with different names but identical FCG structures. Most of the previous datasets primarily consists of samples collected before 2017, potentially limiting its generalizability to modern malware, and hence misleads the ongoing research on malware detection. Additionally, existing malware datasets often include numerous smaller-sized APKs, which limits their utility in comprehensive malware analysis. Moreover, these datasets often lack essential APK properties, such as detailed information on the services or libraries used by the app, which impedes a thorough understanding of the app's behavior and functionality, making it difficult to accurately classify malware. To address these limitations, we ensured that BCG has four key features: (1) larger size to facilitate more robust graph classification, (2) recent data (including 2017 and after) to reflect evolving threats, (3) unique APKs to ensure a more accurate evaluation testbed, and (4) non-graph APK features (graph attributes) for a more holistic evaluation. The summary of our datasets, BCG, is given in Table 2 and here, we briefly describe each of the four features:

## 4.1 LARGER IN SIZE

Existing datasets for analyzing and differentiating APKs through graphs are often too small. While MalNet has the largest dataset size, it contains many graphs with very few nodes/edges. Out of 100K APKs in MalNet, approximately 3,000 FCGs contain less than 100 nodes/edges. Analyzing such small graphs offers limited value for malware classification, or broader graph classification, and may even mislead the classifier to prioritize size. We address this limitation by creating a new dataset specifically designed for graph-based analysis. For this reason, we consider only those with a minimum size of 4MB during the APK collection phase. Additionally, we exclude any graphs with fewer than 100 edges. This ensures our dataset consists of larger graphs, providing more meaningful insights for malware classification as well as type/family classifications within malware.

## 4.2 RECENT AND MODERN APKS

All the existing FCG datasets contain mostly obsolete APKs. For instance, approximately 99% of the malware APKs in the MalNet dataset originate from before 2017, with the remaining 1% are from post-2017. Except that, between 2017 and 2021, the dataset contains only benign APKs. The Android ecosystem has evolved significantly since 2016. Older APKs, built for simpler Android versions, might not reflect the complexities of modern malware threats which can limit the effectiveness of malware detection methods. To address these limitations, we focus on constructing a new malware dataset that incorporates recent APKs. We have collected malware samples specifically targeting those published in 2017 or later. The distribution of APKs across different years is visualized in the histogram of Fig. 2. It is evident that our dataset primarily contains recent APKs, with a significant portion dating from after 2020.

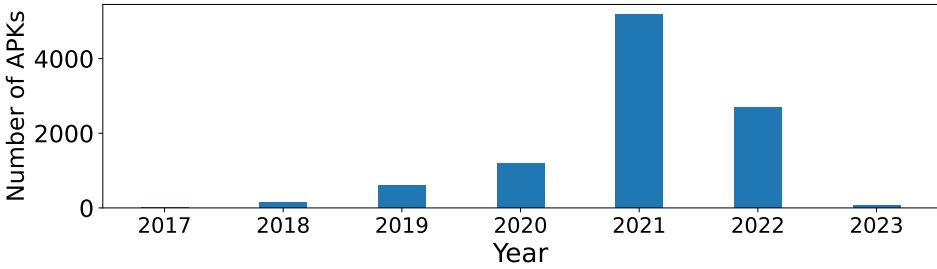

Figure 2: Temporal distribution of APKs in BCG.

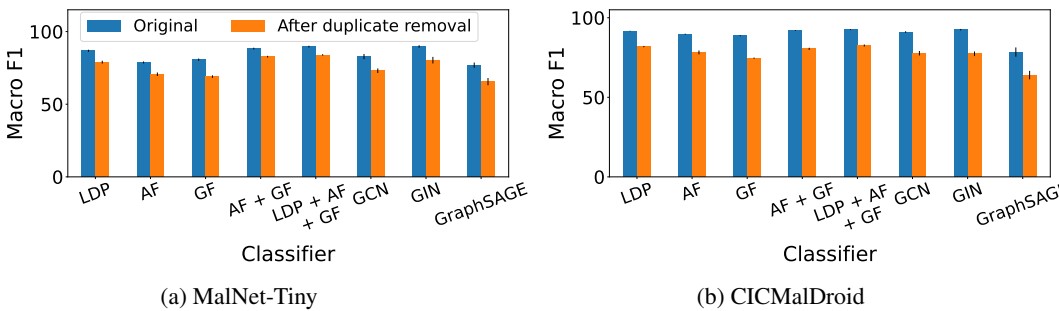

(a) MalNet-Tiny

(b) CICMalDroid

Figure 3: Performance comparison across all methods of MalNet-Tiny and CICMalDroid datasets before and after removing the duplicate APKs. Removing duplicates decreases the performance drastically for all methods.

## 4.3 UNIQUE APKS

High malware classification performance reported by recent approaches might be misleading due to the presence of duplicate APKs in commonly used datasets (Shirzad et al., 2023; Cao et al., 2023; Rampášek et al., 2022). The duplicate APKs have identical FCG structures but appear under different names. This is caused by repackaging, where the same malware is redistributed with minor modifications like altered app icons or backgrounds, or by the same APK being uploaded to multiple app stores. For instance, Malnet-Tiny and CICMalDroid suffer from this significantly. Malnet-Tiny contains around 2,000 repackaged APKs (approximately 40% of the entire data) while CICMalDroid has 41% of duplicate APKs. We have also investigated a subset of the MalNet, 100K out of 1.2M, and observed that approximately 51% of the APKs are duplicates. Importantly, removing these duplicates often can lead to a significant drop in the classification performance as the duplicate FCGs cause **label leakage**, or even database lookups, in the original data when both train and test splits contain the same FCG. We conducted experiments on the original MalNet-Tiny and CICMalDroid, as well as the filtered version after duplicates are removed, using different classifiers (details are in Section 5.2). Figure 3 presents the results. Macro F1 scores decrease after removing the duplicate APKs, consistently for all the classifiers, reaching up to 16.38% decrease in CICMalDroid dataset on GIN method. Motivated by this, we focus on constructing new malware classification datasets that contain unique APKs.

Table 3: List of non-graph APK features and their descriptions.

| Feature | Description |
|---|---|
| APK size | The size of the APK file in bytes. |
| Dex size | The size of the Dex file in bytes. |
| App name | The application name of the APK. |
| Package name | The unique package name of the APK. |
| App permission | The list of permissions requested by the app, indicating the resources and data the app needs access to. |
| App main activity | The main activity of the app and entry point when users launch the app. |
| App all activity | The complete list of all activities defined in the app, representing the different screens and interactions available within the app. |
| Services | The list of all services used by the app, which are components that run in the background to perform long-running operations. |
| Receivers | The list of all broadcast receivers in the app, which are components that respond to system-wide broadcast announcements. |
| Libraries | The list of all libraries used by the app, which can include third-party libraries that provide additional functionality and support. |

## 4.4 Non-graph APK Features (AF)

Prior research (Wang et al., 2020; Lee et al., 2019; Alzaylaee et al., 2020) highlights the effectiveness of basic APK features for malware classification. However, existing datasets do not include these APK features, requiring manual extraction from the APKs. To address this, we have incorporated APK features into our BCG dataset. We extracted two basic features from the APK and manifest files: APK size and DEX size. APK size refers to the entire APK file size in bytes, while DEX size represents the size of the Dalvik Executable (DEX) file also in bytes. The DEX file contains the optimized machine code used by the Android system to run the application. Beyond basic size information, we utilize Androguard (Desnos & Gueguen, 2011) to extract various textual features from the APK, which have been shown to be useful for malware classification in previous works (Wang et al., 2020; Lee et al., 2019). These textual features include the app/package name, permissions requested by the app, all activities of the app, services or libraries used by the app, and the list of broadcast receivers. We encoded all of the textual features using a 100-dimensional TensorFlow sentence encoder (Cer et al., 2018) and further reduced its dimensionality to 2 using t-SNE (Van der Maaten & Hinton, 2008) for efficient processing. t-SNE effectively preserves local structure and retains original information and widely adopted in prior malware detection studies (Yumlembam et al., 2022; Zhu et al., 2018). Therefore, we used t-SNE to obtain two-dimensional features. A detailed description of all non-graph APK features is provided in Table 3.

## 5 Experimental evaluation

In this section, we evaluate the baseline performance of our datasets using various established methods. Initially, we present the experimental setup and then we evaluate all the approaches. The experiments were conducted on a Linux operating system (v. 3.10.0-1127) running on a machine with Intel(R) Xeon(R) Gold 6130 CPU processor at 2.10 GHz with 192 GB memory. An Nvidia A100 GPU was used specifically for the GNN experiments. **Our code is publicly available at** `https://anonymous.4open.science/r/BCG-code/.`

### 5.1 Experimental setup

To assess the graph classification performance of each model on a given dataset (i.e., set of graphs), we employ 70/10/20 train/validation/test split. We utilize macro-F1 score as the primary evaluation metric, considering the imbalanced nature of malware datasets, and also report accuracy, precision, and recall. We repeat each experiment ten times with different random seeds and give the average and standard deviation of these runs.

### 5.2 Graph classification baselines

We consider several state-of-the-art methods for classifying FCGs. These methods encompass both feature-based approaches that analyze characteristics like permissions or app size directly from the APK, and graph-based approaches that focus on the FCG structure and potentially incorporate node features for richer information. Here we summarize each briefly:

**1. Deep learning techniques on graphs:** We consider three established methods based on GNNs and node embeddings: GCN (Kipf & Welling, 2016), GIN (Xu et al., 2018), and LDP (Cai & Wang, 2018). These methods, originally used for malware type and family classification on MalNet, are adapted to our setting. LDP is a simple node representation scheme that summarizes each node and its immediate neighbors using five degree statistics. These features are then aggregated and combined into feature vectors for the GNNs. To ensure consistency, we adopt the same experimental setup in MalNet (Freitas et al., 2020), using 5 GNN layers, Adam optimizer, 64 hidden units, a learning rate of 0.0001, and LDP node features for GCN and GIN. We also incorporated GraphSAGE into our evaluation alongside GCN and GIN given its established effectiveness for malware classification (Lo et al., 2022; Yumlembam et al., 2022; Vinayaka & Jaidhar, 2021). Like GCN and GIN, GraphSAGE was implemented following the same experimental setup for a consistent comparison.

**2. Random forest on app-level features:** To investigate the effectiveness of basic APK properties, that are unrelated to FCGs, we extracted various features from the original APK files by using Androguard (details are discussed in Section 4.4). We refer to these as APK Features, AF for short.

Table 4: Malware type classification results using different approaches on three datasets: MalNet-Tiny (MT), CICMaldroid (CMD), and our dataset BCG. MT* and CMD* indicate the results after eliminating duplicates. Best scores are marked in bold for each dataset and method.

| Method | Accuracy | | | | | Macro-F1 | | | | |
|---|---|---|---|---|---|---|---|---|---|---|
| | MT | MT* | CMD | CMD* | BCG | MT | MT* | CMD | CMD* | BCG |
| LDP | 86.6 ± 0.7 | 77.9 ± 1.0 | 92.2 ± 0.1 | 87.3 ± 0.3 | 70.2 ± 0.4 | 86.7 ± 0.7 | 78.9 ± 1.0 | 91.6 ± 0.1 | 81.8 ± 0.3 | 17.9 ± 0.5 |
| AF | 78.8 ± 0.6 | 74.8 ± 1.0 | 90.8 ± 0.2 | 85.0 ± 0.8 | 70.4 ± 0.2 | 78.7 ± 0.6 | 70.6 ± 1.2 | 89.7 ± 0.0 | 78.1 ± 1.2 | 15.6 ± 0.9 |
| GF | 80.5 ± 0.7 | 66.8 ± 0.8 | 89.9 ± 0.2 | 82.3 ± 0.1 | 63.8 ± 0.4 | 80.6 ± 0.7 | 69.0 ± 0.8 | 88.6 ± 0.2 | 74.6 ± 0.3 | 12.7 ± 0.5 |
| AF+GF | 88.3 ± 0.0 | 82.9 ± 0.5 | 92.6 ± 0.2 | 86.9 ± 0.4 | 72.2 ± 0.2 | 88.3 ± 0.0 | 82.7 ± 0.5 | 91.9 ± 0.2 | 80.4 ± 0.6 | 16.8 ± 0.8 |
| LDP+AF+GF | 89.6 ± 0.5 | **83.8 ± 0.5** | **93.0 ± 0.2** | **88.1 ± 0.3** | **73.2 ± 0.2** | 89.6 ± 0.6 | **83.9 ± 0.0** | **92.5 ± 0.2** | **82.5 ± 0.6** | **19.3 ± 1.0** |
| GCN | 82.8 ± 1.0 | 73.2 ± 2.2 | 91.3 ± 0.4 | 84.9 ± 0.9 | 66.5 ± 4.8 | 82.8 ± 1.7 | 73.0 ± 1.5 | 91.0 ± 0.0 | 77.7 ± 1.4 | 14.1 ± 3.5 |
| GIN | **89.7 ± 0.0** | 80.8 ± 1.9 | 92.7 ± 0.4 | 85.1 ± 0.5 | 66.9 ± 1.6 | **89.7 ± 0.8** | 80.3 ± 2.2 | 92.5 ± 0.4 | 77.3 ± 1.4 | 12.9 ± 2.1 |
| GraphSAGE | 76.5 ± 2.0 | 65.1 ± 2.2 | 79.1 ± 2.6 | 75.3 ± 2.4 | 48.7 ± 5.6 | 76.8 ± 1.8 | 65.5 ± 2.4 | 78.4 ± 2.8 | 63.9 ± 2.6 | 4.54 ± 0.8 |

| Method | Precision | | | | | Recall | | | | |
|---|---|---|---|---|---|---|---|---|---|---|
| | MT | MT* | CMD | CMD* | BCG | MT | MT* | CMD | CMD* | BCG |
| LDP | 87.4 ± 0.6 | 80.9 ± 0.8 | 91.3 ± 0.1 | 82.8 ± 0.5 | 24.4 ± 0.9 | 86.6 ± 0.7 | 78.6 ± 0.9 | 92.1 ± 0.3 | **82.1 ± 0.4** | 15.6 ± 0.4 |
| AF | 79.8 ± 0.5 | 75.5 ± 2.0 | 89.2 ± 0.2 | 80.2 ± 1.0 | 26.3 ± 2.0 | 78.8 ± 0.6 | 69.4 ± 1.0 | 90.4 ± 0.3 | 78.0 ± 1.1 | 13.3 ± 0.5 |
| GF | 81.1 ± 0.7 | 69.7 ± 1.1 | 88.5 ± 0.2 | 75.1 ± 0.0 | 14.8 ± 1.3 | 80.5 ± 0.7 | 69.5 ± 0.7 | 88.8 ± 0.3 | 74.3 ± 0.3 | 12.6 ± 0.3 |
| AF+GF | 88.7 ± 0.5 | 83.9 ± 0.4 | 91.5 ± 0.2 | 82.4 ± 0.7 | 24.6 ± 1.7 | 88.3 ± 0.0 | 82.6 ± 0.6 | 92.5 ± 0.2 | 79.9 ± 0.6 | 14.7 ± 0.6 |
| LDP+AF+GF | **90.1 ± 0.5** | **85.1 ± 0.0** | 92.1 ± 0.3 | **84.1 ± 0.7** | **27.7 ± 1.8** | 89.6 ± 0.5 | **83.9 ± 0.4** | **93.0 ± 0.2** | 82.1 ± 0.5 | **16.9 ± 0.7** |
| GCN | 83.3 ± 1.4 | 72.8 ± 1.5 | 91.1 ± 0.5 | 77.4 ± 1.6 | 17.3 ± 4.6 | 82.8 ± 1.0 | 74.7 ± 1.4 | 90.9 ± 0.5 | 78.3 ± 1.3 | 14.3 ± 3.0 |
| GIN | 90.0 ± 0.7 | 80.6 ± 2.2 | **92.4 ± 0.5** | 77.8 ± 0.0 | 16.5 ± 3.1 | **89.7 ± 0.0** | 81.2 ± 1.8 | 92.6 ± 0.4 | 77.6 ± 1.4 | 13.9 ± 2.6 |
| GraphSAGE | 78.0 ± 1.6 | 65.2 ± 2.3 | 78.5 ± 2.6 | 64.3 ± 2.6 | 4.55 ± 0.6 | 76.5 ± 2.0 | 67.5 ± 2.2 | 79.6 ± 2.4 | 64.1 ± 2.5 | 5.48 ± 0.9 |

We also constructed graph features, GF, derived from the FCGs, by capturing simple graph analytics, such as the number of nodes/edges, largest connected component size, and centrality metrics, detailed descriptions are given in Table 6 at Appendix. Finally, we combine AF and GF, and feed them into a Random Forest model for malware classification.

**3. Combined approach:** While both FCGs and basic APK features are valuable, recent research suggests that their combined use can lead to even better performance. APK features capture high-level information about the app (permissions, size), while FCGs provide detailed insights into the app's functionality through call relationships between functions. Hence, we explore the effectiveness of combining all app-level features (AF + GF) with LDP node embeddings derived from FCGs. LDP node embeddings are aggregated to create graph-level feature vectors, which are then merged with AF + GF to form a comprehensive feature set. We then evaluate this combined feature set using Random Forest classification model. To optimize hyperparameters like the number of estimators and tree depth, we perform a grid search on the validation set, replicating the configuration used by MalNet for these models.

## 5.3 Performance analysis

**Malware type classification.** We evaluate the classification performance of the aforementioned classifiers on three datasets: MalNet-Tiny, CICMalDroid, and our new BCG dataset. Malware type (in BCG) is one of 29 classes, which is either benign (i.e., not malware) or a specific type of malware, as denoted in Table 2. Table 4 gives the results. All the results on MalNet-Tiny and CICMalDroid are drastically better than those on BCG. For accuracy, the best classifier yields 89.76% and 93.04% on MalNet-Tiny and CICMaldroid, whereas it is only 73.2% on BCG. There is even a more drastic difference in Macro-F1 (and Precision and Recall): the best classifier can easily reach to around 90% on MalNet-Tiny and CICMalDroid but can only yield 19% on BCG! Table 4 also includes the results of MalNet-Tiny and CICMalDroid datasets after removing duplicates. Even after removing duplicates, all the methods achieve higher performance on these datasets than BCG. Notably, the lowest Macro-F1 score (65.5) on MalNet-Tiny with GraphSAGE is significantly higher than the BCG equivalent (4.54). This highlights the inherent difficulty of classifying malware in BCG.

From those results, it is evident that state-of-the-art approaches can easily obtain high performance for malware classification on previously curated datasets, however the same cannot be said for our newly and carefully constructed BCG dataset. The state-of-the-art methods fail miserably on identifying malwares on BCG. This indicates that malware classification using FCGs is significantly more complex than previously thought. There is a clear need for new graph classification techniques that can handle the complexities within the BCG.

Table 5: Malware family classification results on BCG dataset using different approaches.

| Method | Accuracy | Macro-F1 | Precision | Recall |
|---|---|---|---|---|
| LDP | $77.22 \pm 0.23$ | $19.42 \pm 0.88$ | $28.12 \pm 1.3$ | $16.81 \pm 0.79$ |
| APK features (AF) | $75.53 \pm 0.24$ | $14.31 \pm 0.83$ | $26.26 \pm 1.17$ | $11.88 \pm 0.75$ |
| Graph features (GF) | $73.87 \pm 0.28$ | $13.59 \pm 0.29$ | $18.14 \pm 0.69$ | $12.3 \pm 0.16$ |
| AF + GF | $\mathbf{79.34 \pm 0.22}$ | $19.02 \pm 0.34$ | $30.67 \pm 0.66$ | $16.27 \pm 0.27$ |
| LDP + AF + GF | $79.3 \pm 0.34$ | $\mathbf{20.85 \pm 1.05}$ | $\mathbf{32.46 \pm 1.43}$ | $\mathbf{17.59 \pm 0.94}$ |
| GCN | $73.68 \pm 1.91$ | $7.3 \pm 3.32$ | $9.11 \pm 4.53$ | $6.99 \pm 3.21$ |
| GIN | $71.07 \pm 1.22$ | $2.83 \pm 1.0$ | $3.51 \pm 1.43$ | $2.71 \pm 0.87$ |
| GraphSAGE | $67.33 \pm 0.04$ | $0.66 \pm 0.02$ | $0.62 \pm 0.14$ | $0.81 \pm 0.01$ |

**Malware family classification.** Beyond malware type classification, our BCG dataset also include 118 family labels over all APKs, enabling a more granular analysis. We conducted experiments to classify malware according to their families, results are in Table 5. Interestingly, traditional GNNs like GraphSAGE (Macro-F1 < 1%) and GIN (Macro-F1 = 2.83%) achieved low performance on this family classification task within the BCG dataset. This suggests that these models might require further development or additional data augmentation to handle the complexities present in our data.

**Evaluating the difficulty of classifying recent APKs.** The complexity of both benign and malicious applications has significantly evolved in recent years. To empirically validate this, we conduct two sets of experiments. The first experiment involves a temporal split of the data, using earlier data for training and validation, and later data for testing. Such a temporal split results in lower accuracy compared to random data partitioning (details are in Table 9 at Appendix). For the second experiment, we divide the BCG dataset into two equal halves: 2017-June 2021 and July 2021-2023. Each half is independently evaluated with its own training and testing sets. We observe that the second half consistently exhibited lower performance (details are in Table 10 at Appendix). These findings collectively suggest that malware classification becomes increasingly challenging for more recent APKs and including obsolete APKs in an FCG dataset can result in inflated classification performance.

# 6 CONCLUSION

Traditional malware classification datasets struggle with redundancy, limited size, and outdated data. These limitations hinder the development of effective models for detecting modern malware threats. This work addresses these issues by introducing the BCG dataset, a collection of recent and unique FCGs containing benign and malware classes as well as different types and families of malware. In our BCG dataset, comprising 9938 graphs, the average size is 25k nodes and 54k edges. It spans a diverse hierarchy of 29 types and 118 families. The analysis of BCG dataset revealed promising avenues for future research. By overcoming the limitations of existing datasets, BCG paves the way for significant advancements in malware classification research.

**Limitations.** While the BCG dataset is valuable for FCG-based malware classification, it does not encompass other aspects of malware analysis, such as identifying malicious code or unpacking obfuscated content. Future research could explore expanding BCG to include these attributes for a more comprehensive analysis.

**Reproducibility Statement:** All of our experimental results are reproducible. The anonymous code for replicating the baseline results is publicly available at `https://anonymous.4open.science/r/BCG-code/`, with further details provided in Section 5. The dataset and its descriptions can be accessed at `https://iclr.me/`.

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

# A APPENDIX

## A.1 DESCRIPTION OF GRAPH FEATURES (GF)

Table 6: List of graph features and their descriptions.

| Feature | Description |
|---|---|
| Num Nodes | The number of nodes in the Function Call Graph. |
| Num Edges | The number of edges in the Function Call Graph. |
| Node degree | The degree of the nodes in the Function Call Graph. |
| Selfloop | The number of self-loops in the Function Call Graph. |
| Indegree | The indegree of the nodes in the Function Call Graph. |
| Closeness | The closeness centrality of the nodes in the Function Call Graph. |
| Num Cycle | The number of cycles in the Function Call Graph. |
| Large Conn | The size of the largest connected component in the Function Call Graph. |
| Large Conn Ratio | The ratio of the size of the largest connected component to the total number of nodes in the graph. |
| Large Weak conn | The size of the largest weakly connected component in the Function Call Graph. |
| Large Weak conn Ratio | The ratio of the size of the largest weakly connected component to the total number of nodes. |
| Second Large Weak Conn | The size of the second largest weakly connected component in the Function Call Graph. |
| Second Large Weak Conn Ratio | The ratio of the size of the second largest weakly connected component to the total number of nodes. |
| Power Alpha | The alpha parameter of the power-law distribution fitted to the node degrees. |
| Power Sigma | The sigma parameter of the power-law distribution fitted to the node degrees. |

Table 7: Distribution of BCG types across the years 2017 to 2023.

| Type/Year | 2017 | 2018 | 2019 | 2020 | 2021 | 2022 | 2023 |
|---|---|---|---|---|---|---|---|
| benign | 25 | 11 | 17 | 64 | 4077 | 1608 | 78 |
| trojan | 2 | 35 | 200 | 414 | 439 | 434 | 1 |
| adware | 5 | 49 | 120 | 187 | 204 | 238 | 0 |
| smsreg | 0 | 2 | 50 | 108 | 70 | 49 | 0 |
| adware++trojan | 0 | 3 | 51 | 69 | 59 | 68 | 0 |
| riskware | 3 | 13 | 30 | 23 | 35 | 53 | 0 |
| smsreg++trojan | 0 | 1 | 24 | 42 | 36 | 18 | 0 |
| riskware++trojan | 1 | 8 | 4 | 30 | 43 | 33 | 0 |
| risktool++trojan | 0 | 5 | 29 | 20 | 34 | 26 | 0 |
| addisplay | 0 | 6 | 17 | 24 | 21 | 14 | 0 |
| dropper++trojan | 0 | 0 | 2 | 35 | 23 | 14 | 0 |
| risktool | 0 | 0 | 10 | 23 | 18 | 16 | 0 |
| spy++trojan | 1 | 3 | 5 | 12 | 19 | 17 | 0 |
| banker++trojan | 0 | 0 | 2 | 34 | 12 | 6 | 0 |
| adware++riskware | 0 | 2 | 6 | 12 | 8 | 19 | 0 |
| risktool++riskware | 0 | 2 | 4 | 13 | 15 | 13 | 0 |
| spr++trojan | 1 | 0 | 11 | 8 | 7 | 8 | 0 |
| riskware++smsreg | 3 | 4 | 0 | 6 | 13 | 4 | 0 |
| rog | 0 | 2 | 3 | 12 | 7 | 6 | 0 |
| smsreg++spr | 0 | 0 | 8 | 7 | 6 | 9 | 0 |
| downloader++trojan | 0 | 0 | 4 | 7 | 10 | 8 | 0 |
| spy | 0 | 2 | 2 | 5 | 9 | 9 | 0 |
| clicker++trojan | 0 | 0 | 1 | 10 | 13 | 2 | 0 |
| fakeapp++trojan | 1 | 0 | 1 | 5 | 5 | 1 | 0 |
| risktool++spr | 0 | 0 | 3 | 5 | 3 | 2 | 0 |
| fakeapp | 0 | 1 | 0 | 1 | 3 | 3 | 0 |
| adware++risktool | 1 | 0 | 2 | 3 | 0 | 1 | 0 |
| backdoor | 0 | 1 | 2 | 0 | 1 | 3 | 0 |
| backdoor++trojan | 0 | 0 | 1 | 1 | 2 | 3 | 0 |

Table 8: Distribution of BCG family labels (top 15) across years 2017 to 2023.

| Family/Year | 2017 | 2018 | 2019 | 2020 | 2021 | 2022 | 2023 |
|---|---|---|---|---|---|---|---|
| artemis | 3 | 24 | 124 | 302 | 335 | 237 | 0 |
| jiagu | 0 | 6 | 56 | 79 | 95 | 78 | 1 |
| kuguo | 1 | 11 | 43 | 50 | 54 | 24 | 0 |
| smspay | 4 | 9 | 30 | 34 | 28 | 39 | 0 |
| gexin | 1 | 12 | 22 | 17 | 20 | 42 | 0 |
| tencent | 0 | 1 | 24 | 21 | 24 | 22 | 0 |
| bankbot | 0 | 0 | 0 | 36 | 36 | 14 | 0 |
| jpush | 0 | 1 | 14 | 14 | 15 | 40 | 0 |
| malct | 0 | 0 | 11 | 20 | 24 | 27 | 0 |
| dowgin | 2 | 0 | 13 | 8 | 25 | 27 | 0 |
| kyview | 2 | 2 | 7 | 30 | 12 | 18 | 0 |
| autoins | 0 | 3 | 10 | 14 | 20 | 23 | 0 |
| triada | 1 | 2 | 8 | 15 | 18 | 21 | 0 |
| ads | 0 | 2 | 9 | 15 | 20 | 15 | 0 |
| secneo | 0 | 2 | 5 | 21 | 14 | 17 | 0 |

Table 9: Impact of time-based data split on accuracy compared to random data partitioning, with earlier data used for training and validation, and later data for testing.

| Method | Accuracy | | Macro-F1 | |
|---|---|---|---|---|
| | Random split | Time-based split | Random split | Time-based split |
| LDP | 68.14 ± 1.51 | **62.69 ± 0.38** | 16.05 ± 2.62 | **13.44 ± 0.72** |
| AF | 70.43 ± 0.25 | **63.78 ± 0.36** | 15.6 ± 0.91 | **10.29 ± 0.39** |
| GF | 63.83 ± 0.43 | **55.71 ± 0.45** | 12.7 ± 0.54 | **9.02 ± 0.63** |
| AF + GF | 66.1 ± 0.67 | **60.74 ± 0.24** | 15.34 ± 2.3 | **12.43 ± 0.48** |
| LDP + AF + GF | 68.34 ± 0.92 | **65.62 ± 0.28** | 16.12 ± 1.23 | **11.88 ± 0.72** |
| | Precision | | Recall | |
| | Random split | Time-based split | Random split | Time-based split |
| LDP | 22.21 ± 2.7 | **21.1 ± 1.51** | 14.53 ± 2.6 | **11.76 ± 0.49** |
| AF | 26.3 ± 2.08 | **15.77 ± 1.11** | 13.36 ± 0.58 | **9.83 ± 0.22** |
| GF | 14.86 ± 1.38 | **11.0 ± 1.35** | 12.68 ± 0.39 | **8.88 ± 0.76** |
| AF + GF | 22.05 ± 1.2 | **17.26 ± 1.05** | 13.05 ± 2.3 | **11.12 ± 0.28** |
| LDP + AF + GF | 21 ± 0.62 | **18.69 ± 1.1** | 14.76 ± 2.9 | **10.77 ± 0.55** |

Table 10: Performance comparison between the first and second halves of the BCG dataset, with independent evaluation using separate training and testing sets.

| Method | Accuracy | | Macro-F1 | |
|---|---|---|---|---|
| | BCG first half | BCG second half | BCG first half | BCG second half |
| LDP | 72.06 ± 0.32 | **59.43 ± 0.43** | 15.02 ± 0.78 | **10.29 ± 1.72** |
| AF | 77.9 ± 0.4 | **62.26 ± 0.42** | 13.42 ± 2.42 | **7.71 ± 0.43** |
| GF | 67.09 ± 2.32 | **56.43 ± 0.3** | 11.24 ± 0.81 | **8.87 ± 0.52** |
| AF + GF | 65.39 ± 0.56 | **62.07 ± 0.32** | 14.49 ± 1.58 | **7.89 ± 0.14** |
| LDP + AF + GF | 75.05 ± 0.31 | **63.23 ± 0.45** | 16.07 ± 0.52 | **9.56 ± 0.22** |
| | Precision | | Recall | |
| | BCG first half | BCG second half | BCG first half | BCG second half |
| LDP | 16.42 ± 1.04 | **13.78 ± 1.94** | 16.67 ± 0.55 | **9.65 ± 1.73** |
| AF | 14.32 ± 2.82 | **10.39 ± 1.49** | 14.73 ± 2.79 | **7.45 ± 0.36** |
| GF | 10.3 ± 1.04 | **9.07 ± 0.73** | 14.13 ± 1.11 | **10.32 ± 0.17** |
| AF + GF | 17.05 ± 3.36 | **9.14 ± 0.39** | 15.98 ± 1.55 | **8.0 ± 0.12** |
| LDP + AF + GF | 16.46 ± 0.53 | **13.09 ± 1.18** | 17.4 ± 0.54 | **8.98 ± 0.17** |

