# OpenReview forum: "Better Call Graphs: A New Dataset of Function Call Graphs for Malware Classification"
_ICLR.cc/2025/Conference — ICLR 2025 Conference Withdrawn Submission_

### Official Review · Reviewer_Mon5 · 2024-10-31

**Soundness:** 3
**Presentation:** 3
**Contribution:** 3
**Rating:** 3
**Confidence:** 3

**Summary:**

This paper proposes a new comprehensive dataset, Better Call Graphs (BCG), that contains large and unique FCGs from recent APKs,
along with graph-level APK features, with benign and malware samples from different types and families. The BCG datasets and existing datasets are evaluated on several baseline approaches to demonstrate the necessity of the BCG dataset.

**Strengths:**

This paper addresses the limitations of existing malware classification datasets, particularly the issue of outdated malware samples. A new, open-source dataset has been created, covering malware samples from 2017 to 2023.

**Weaknesses:**

First, the primary contribution of this work is the creation of a new dataset covering recent malware samples. However, I do not think this contribution meets the expectation of ICLR. The creation of a new dataset is largely engineering efforts and lacks sufficient scientific contribution.

Second, the evaluation shows that baseline methods perform poorly on this newly created dataset. I would appreciate the author to design a new model that can work well on the new dataset to enhance the scientific contribution.

Third, I find the poor performance of the baseline methods somewhat unclear. Could the authors clarify the experimental setup? Specifically, was the baseline trained on a portion of the new dataset (e.g., 70%) and then tested on the remaining samples?

Finally, the new dataset spans malware samples from 2017 to 2023, yet there are several other datasets that include samples up to 2021. Do the samples in your dataset overlap with those in existing datasets? If overlaps exist, would collecting malware samples only from 2021 to 2023 have been sufficient?

**Questions:**

1. Clarify the contribution of this work, especially scientific contribution.
2. Explain the experimental setup: whether baseline models are trained on the new created dataset?
3. Clarify the overlap between the new created datasets and existing datasets (especially those includes samples up to 2021).

---

> ### Author Response · Authors · 2024-11-24
>
> Thank you for your comments. First, we address the Weaknesses you mentioned.
>
>
> **W1**: We respectfully disagree that creation of a new dataset lacks scientific contribution. ICLR 2025 Call for Papers explicitly mentions “datasets and benchmarks” as a relevant topic. Our BCG dataset addresses critical limitations in existing datasets that have led to misleading results in malware detection research. Here we summarize the key scientific contributions of our work:
>
> - We introduce a novel dataset (BCG) that addresses critical limitations in existing malware detection datasets, particularly the presence of duplicates and outdated samples.
>
> - We demonstrate empirically how duplicate samples in existing datasets lead to inflated performance metrics, potentially misleading researchers about the effectiveness of their methods. This finding has significant implications in this field.
>
> - By providing unique, recent, and larger FCGs along with non-graph features, BCG facilitates more robust and realistic evaluations of malware detection methods. This enables researchers to design and test approaches that address current malware threats effectively.
>
> - The dataset provides a benchmark for evaluating the effectiveness of malware analysis techniques and inspires novel algorithmic development. We believe that this contribution is essential for advancing the field and ensuring that reported performances reflect real-world scenarios.
>
>
> **W2**: We appreciate the suggestion to design a new model that works well on our dataset. However, our primary goal with this paper is to establish a robust benchmark dataset and demonstrate its necessity. The poor performance of baseline models highlights the challenges posed by our dataset, encouraging the development of advanced methods.
>
>
> **W3**: In our experiments, the baseline models were trained on 70% of the new dataset, validated on 10%, and tested on the remaining 20% (see Lines 407-409 in the revised pdf). Detailed descriptions and results can be found in Section 5. This setup follows the approach used in the MalNet (Freitas et al., 2020) paper. The poor performance of the baseline models reflects the unique characteristics of our dataset, which includes non-duplicate and recent APKs absent in other datasets. This performance highlights the dataset's unique challenges and realism, making it a valuable resource for advancing the field.
>
>
> **W4**: Table 1 provides a brief comparison of previous Android-based FCG datasets and BCG. Among these, only MalNet (and its tiny version) includes APKs from 2017 onward, with family information. However, just 1% of MalNet's samples are from 2017, and all are benign. In contrast, our dataset spans 2017 to 2023, includes both benign and malignant APKs, and provides family information. Furthermore, unlike MalNet and other datasets, our dataset includes non-graph features and doesn't contain any duplicate sample.
>
>
> Now, we address the questions you asked,
>
> **Q1**: **Please refer to our answer to W1 above.**
>
>
> **Q2**: Experimental setup and detailed results are provided in Section 5. **Also, please see our comments on W3 above.**
>
>
> **Q3**: Detailed comparison between the new created datasets and existing datasets are provided in section 2.2.  **Please see our comments on W4 above for further details.**
>
> We sincerely thank the reviewer again for the detailed feedback and are glad to address any additional questions or concerns.

---

> > ### Author Response · Authors · 2024-12-01
> >
> > Thanks again for your valuable feedback. Could you please acknowledge that you read our response to your comments? If your concerns are addressed, we’d be grateful if you can adjust the score. If not, we’d love to engage further to address your comments.

---

### Official Review · Reviewer_zMcY · 2024-11-02

**Soundness:** 2
**Presentation:** 3
**Contribution:** 1
**Rating:** 5
**Confidence:** 4

**Summary:**

The authors propose a new comprehensive datasetthat contains large and unique FCGs from recent APKs, along with graph-level APK features.

**Strengths:**

1. A new dataset is proposed.
2. The paper is well organized.

**Weaknesses:**

The collection and construction of the dataset lack distinctiveness. The inclusion of new software and non-repetitive samples does not constitute the primary contribution of the paper.

**Questions:**

1. I am not entirely sure why the authors refer to their dataset as the FCG dataset. Other FCG datasets mentioned by the authors, such as Drebin and CIC, provide SHA256 hashes or APK files, which are not directly related to FCG. Additionally, the Drebin dataset paper introduces a binary feature, not an FCG feature. Therefore, I am unsure why the authors classify their dataset as an FCG dataset.
2. The dataset relies heavily on VirusTotal's labeling. Although VirusTotal labeling remains the mainstream method, the authors mention, 'To ensure reliability, we only consider the APKs flagged by multiple antivirus engines in VirusTotal.' How many are considered 'multiple'? Which ones are they? Why were these specific ones chosen?
3. When labeling families using AVClass, there is significant noise. How should samples that cannot be consistently labeled with a family tag be handled?
4. The distribution of year categories is highly imbalanced. Why is this the case? It appears that the sample count is highest for the year 2021.
5. It is hoped that the authors can include the results of common detection methods for function call graph such as Mamadroid and APIGraph on BCG.

[1] Enhancing State-of-the-art Classifiers with API Semantics to Detect Evolved Android Malware
[2] MAMADROID: Detecting Android Malware by Building Markov Chains of Behavioral Models


6. Similarly, when evaluating the detection performance for family classification, I anticipate the detection results of common malicious family classification methods (FalDroid [3] and MDMC [4]), rather than results based on common graph processing methods.

[3] Android Malware Familial Classification and Representative Sample Selection via Frequent Subgraph Analysis
[4] Byte-level malware classification based on Markov images and deep learning.

7. It is unclear whether the authors have provided temporal information for each APK. This would aid readers in conducting research on concept drift ([5]). Additionally, the authors could conduct additional experiments on concept drift, such as investigating how the model classification accuracy decreases with time on this dataset.

[5]Transcending Transcend: Revisiting Malware Classification in the Presence of Concept Drift


Some related papers that I think need to be studied：
MalRadar: Demystifying Android Malware in the New Era

---

> ### Author Response · Authors · 2024-11-24
>
> Thank you for your comments. First, we address the Weakness you mentioned.
>
> We believe that our data collection and construction methodology is novel as it yields a challenging dataset for Android-based malware classification. We show that existing datasets often yield misleading scores when state-of-the-art classifiers are applied. We emphasize that the inclusion of new software and non-repetitive samples is not the sole contribution of this paper. Our main contribution is the development of a high-quality, up-to-date dataset that addresses critical gaps in existing malware datasets, including duplicates and outdated samples. The focus of our work is on dataset construction, providing a solid foundation for future research and enabling the development of more effective malware detection techniques.
>
> Now, we address the questions you listed,
>
> **Q1**: We would like to clarify that the term "FCG" (Function Call Graph) for our dataset refers to the use of function call graphs as a central feature for malware analysis. While the original Drebin and CIC datasets include SHA256 hashes or APK files, subsequent research has utilized these datasets (SHA256 or APK files) to construct FCGs and analyze performance based on those FCGs. The following works have utilized these datasets for FCG-based analysis:
>
> - [1] Yumlembam et al. (2022), Vinayaka & Jaidhar (2021), Feng et al. (2020), Zhu et al. (2018).
> - [2] Ge, Xiuting, et al. "AMDroid: android malware detection using function call graphs."
> - [3] Yang, Yang, et al. "Android malware detection based on structural features of the function call graph."
>
> **Q2**: We did not prioritize any specific antivirus engines within VirusTotal. For reliability, we considered an APK as malware if it was flagged by more than two antivirus engines, regardless of which engines they were. This threshold was chosen because requiring multiple detections helps reduce the likelihood of false positives from individual engines. We did not give preference to any specific antivirus to avoid introducing bias and to leverage the diverse detection strategies used by different engines to enhance the robustness.
>
>
> **Q3**: AVClass is effective in categorizing malware into families and type/classes. However, for samples that cannot be consistently labeled with a specific family, we addressed this by focusing on the broader class or type of malware (e.g., ransomware, trojans) rather than the exact family name. This approach ensures that the classification remains meaningful and reduces the impact of noise in downstream analyses, enabling reliable results for types or classes despite variability in family tagging.
>
> **Q4**: We downloaded the APKs from AndroZoo and VirusShare. The imbalance in year distribution, particularly for 2021, is due to increased malware additions to these repositories and higher malware production and detection that year. Note that conceptually year distribution has no direct bearing on malware structure. Normalizing over year would unnaturally bias toward specific malware structures. We opted to normalize over structural properties of the malware instead.
>
> **Q5**: Thank you for your suggestions; we now cited these papers in the related work section of the revised pdf. However, recent GNN-based approaches (Yumlembam et al., 2022; Wu et al., 2023; Gunduz, 2022) have shown significant advancements in Android malware detection. Our study focuses on benchmarking results from the latest methods, including those used in MalNet (Freitas et al., 2020), to ensure a more relevant and up-to-date comparison.
>
> **Q6**: Thank you for your suggestions; we now cited these papers in the revised pdf. Given the superior performance of recent GNN-based approaches, we prioritized these newer methods for benchmarking. Our study includes results from several state-of-the-art methods, including those highlighted in MalNet (Freitas et al., 2020), to ensure better comparisons. Instead of focusing on earlier works like FalDroid and MDMC, we prioritized more recent methods that have demonstrated enhanced performance.
>
> **Q7**: Temporal information, including precise date and time for each APK, is provided in the APK feature file available on our dataset website. Note that, we conducted two sets of experiments to examine how model classification accuracy changes over time (see Lines 504-514 in the revised pdf), and the results show a decrease in accuracy.
>
> Thank you for suggesting the MalRadar paper. The MalRadar dataset provides only family-level information and does not include details on the type or category of malware. Regardless, we now cited it in our revised PDF.
>
> Overall, the suggested papers provide useful insights into malware classification. However, our primary contribution is the creation of a high-quality, up-to-date dataset, not an exhaustive comparison of existing methods. We include benchmark results using SOTA methods like those in MalNet, leaving future research to explore other approaches.

---

> > ### Author Response · Authors · 2024-12-01
> >
> > Thanks again for your valuable feedback. Could you please acknowledge that you read our response to your comments? If your concerns are addressed, we’d be grateful if you can adjust the score. If not, we’d love to engage further to address your comments.

---

### Official Review · Reviewer_wSc1 · 2024-11-03

**Soundness:** 2
**Presentation:** 3
**Contribution:** 2
**Rating:** 5
**Confidence:** 4

**Summary:**

The authors propose a dataset for function call graphs (FCGs) from
Android APKs in the task of malware classifcation.  They downloaded
more recent APKs, determined family and type, constructed FCGs, and
removed graphs with fewer than 100 edges and duplicates.  The
resulting dataset has 9938 graphs, with an average of 25k nodes and
54k edges. It contains 29 types and 118 families.

They extracted non-graph APK features (AF), such as servies,
receivers, and libraries.  They also extracted graph features (GF)
such as number of nodes/edges, largest connected component size, and
centrality metrics.  Finally, they extracted node representation based
on LDP (Cai and Wang, 2018) for some of their experiments.

They compare Random Forrest with different combination of features
with 3 GNN algorithms.  Empirical result indicate Random Forest using
all 3 types of features generally outperforms.

**Strengths:**

The authors propose a dataset for function call graphs (FCGs) from
Android APKs in the task of malware classifcation.  They downloaded
more recent APKs, determined family and type, constructed FCGs, and
removed graphs with fewer than 100 edges and duplicates.  The
resulting dataset has 9938 graphs, with an average of 25k nodes and
54k edges. It contains 29 types and 118 families.

They extracted non-graph APK features (AF), such as servies,
receivers, and libraries.  They also extracted graph features (GF)
such as number of nodes/edges, largest connected component size, and
centrality metrics.  Finally, they extracted node representation based
on LDP (Cai and Wang, 2018) for some of their experiments.

They compare Random Forrest with different combination of features
with 3 GNN algorithms.  Empirical result indicate Random Forrest using
all 3 types of features generally outperforms.

**Weaknesses:**

New algorithmic methods were not proposed and evaluated.

More non-graph features could have been extracted, such as n-grams of instructions.

**Questions:**

Why LDP was chosen to generate node embeddings?

The number of nodes across graphs is a variable, how are the node embeddings of a graph converted into a fixed-length feature vector for Random Forest?

---

> ### Author Response · Authors · 2024-11-24
>
> Thank you for your comments. First, we address the Weaknesses you mentioned.
>
>
> **W1**:  Our primary contribution is the construction of a high-quality malware dataset, rather than proposing new algorithmic methods. We focus on providing a solid benchmark for malware analysis using existing techniques. We believe that the dataset itself can serve as a valuable resource for the research community, enabling further exploration of novel algorithmic approaches.
>
> **W2**: Our research specifically aims to address the limitations of existing Android malware datasets and improve their quality. Comparable datasets, such as MalNet and Drebin, also do not include non-graph features. However, our publicly available dataset contains both non-graph features and the APKs themselves. Future researchers can utilize these APKs to extract more non-graph features and analyze their impact on malware classification. Note that n-gram features lose spatial properties of programs, do not capture dataflow, and are easily broken through obfuscation techniques like random NOP injection.
>
>
> Now, we address the question you asked,
>
> **Q1**: We chose LDP to generate node embeddings because it is a well-established and effective method for preserving both local and global graph structures in a low-dimensional space. Additionally, LDP is simple yet often outperforms more complex models and has been used in prior research, including MalNet (Freitas et al., 2020), for graph embedding tasks, making it a reliable choice for our study.
>
> **Q2**: To handle the varying number of nodes across graphs, node embeddings are converted into a fixed-length feature vector using histograms. Each node feature is represented by a histogram with 32 bins, and these histograms are concatenated to form a single fixed-length vector for the entire graph. This histogram-based approach preserves the distributional information of node-level features across the graph, making it suitable for input into a Random Forest model. We adopted this setup from the MalNet (Freitas et al., 2020) paper to ensure consistency.
>
> We sincerely thank the reviewer again for the detailed feedback and are glad to address any additional questions or concerns.

---

> > ### Author Response · Authors · 2024-12-01
> >
> > Thanks again for your valuable feedback. Could you please acknowledge that you read our response to your comments? If your concerns are addressed, we’d be grateful if you can adjust the score. If not, we’d love to engage further to address your comments.

---

> > ### Comment · Reviewer_wSc1 · 2024-12-01
> >
> > I suggest incluidng your response to my questions in your paper.   I'm keeping my rating.

---

### Official Review · Reviewer_U8up · 2024-11-04

**Soundness:** 4
**Presentation:** 4
**Contribution:** 2
**Rating:** 8
**Confidence:** 4

**Summary:**

This paper creates a new dataset for malware. Current Android datasets are outdated and limited in size. Importantly, they often also have duplication, which can be harmful for research. For example, baseline tests on existing datasets highlighted how outdated samples can skew classifier performance. These researchers downloaded malware from AndroZoo and VirusShare, filtered them to find suitable current data, and then used various tools to categorize them.

**Strengths:**

This seems to be a sound way to filter APKs and featurizing them, especially combining graph-level features with the function call graph features. Definitely is a necessary advancement to research, as malware datasets are currently quite old.

**Weaknesses:**

It would be nice to have non-Android (x86) malware.
identifying specific changes in APK structures over time: this is often an important part of malware research - how do these features and graph structures change over time? Similarly, how does this do on unseen data (new families that arise)?
This dataset relies on existing tools like VirusTotal for malware classification and AVClass for label assignment. While I personally don't think this is necessarily an issue, I do think it's relatively a weakness in terms of novelty of methodology.

**Questions:**

Are GIN and GraphSage really the most advanced baselines you have? I think there are malware-specific works that could be useful to this discussion rather than relatively old graph baselines.
What are the statistics on how the classifications/family distrubtions change over time.

---

> ### Author Response · Authors · 2024-11-24
>
> Thank you for your comments. First, we address the Weaknesses you mentioned.
>
> **W1**:  We agree with the suggestion of having non-Android (x86) malware. However, our dataset is primarily focused on Android APKs, given their massive and unique ecosystem, and also widespread usage. Our research specifically aims to address the limitations of existing Android malware datasets and enhance their quality. Regardless, in conclusion we will mention future plans about studying non-Android malware.
>
> **W2**: While we acknowledge that identifying changes in APK structures over time is important, our primary goal is to construct a better dataset to facilitate malware research. However, we have included some relevant information in our paper (see  Lines 153-158 in the revised pdf). Relevant to this, to analyze how model classification accuracy evolves over time, we conducted two sets of experiments (see Lines 504-514 in the revised pdf). The results indicate that malware classification is becoming increasingly challenging. Additionally, while our setup assesses results within known families, future researchers can create splits to explore new family scenarios.
>
> **W3**: Regarding the use of VirusTotal and AVClass, we have chosen these tools because they are well-established and widely recognized for generating malware families and labels. These tools have also been used extensively in prior research (including MalNet), ensuring consistency and reliability in our methodology.
>
> Now, we address the questions you asked,
>
> **Q1**: Although GIN and GraphSAGE are not the newest architectures, they are widely used, well-understood, and are strong baselines for graph tasks. While advanced GNNs like GAT, and VGAE have demonstrated impressive performance, their reliance on external node features (e.g., node2vec, word2vec, pagerank, and other centrality metrics) presents computational challenges and deviates from the core focus of our study. Regardless, we had included GraphSAGE (without external node features) in our evaluation due to its proven effectiveness in malware classification in previous studies (Loet al., 2022; Yumlembam et al., 2022). We anticipate that future research incorporating these architectures and external node features could yield further performance improvements.
>
>
> **Q2**: The APK feature file available on our dataset website contains temporal information (exact date and time) for each APK. Additionally, the type and family distributions for the years 2017–2023 are now added to the Appendix (Table 7 and Table 8) in the revised PDF.
>
> We sincerely thank the reviewer again for the detailed feedback and are glad to address any additional questions or concerns.

---

> > ### Comment · Reviewer_U8up · 2024-12-01
> >
> > Thanks, since my score is high I won't adjust it

---

> > > ### Author Response · Authors · 2024-12-04
> > >
> > > Thank you again for your valuable feedback. We would be grateful if you could champion our work to further support its acceptance.

---

### Author Response · Authors · 2024-12-04

We sincerely thank all the reviewers for their valuable feedback and constructive suggestions. We have addressed all the concerns raised by the reviewers and updated our pdf accordingly (if necessary). While some reviewers have not yet responded to our rebuttals, we are hopeful that our clarifications will be taken into account in the final scores.

---

### Note · Authors · 2025-02-15

I have read and agree with the venue's withdrawal policy on behalf of myself and my co-authors.

---

### Meta-Review · Area_Chair_viYE · 2024-12-15

**Metareview:**

The paper presents a new malware dataset. However, the reviewers expressed concerns that the dataset's collection and construction lack distinctiveness. Both the AC and reviewers agreed that the inclusion of new software and non-repetitive samples alone does not constitute a substantial contribution. To strengthen the paper, it is suggested that the authors explore how features and graph structures evolve over time, as this could better emphasize the significance and utility of the new dataset.

**Additional Comments On Reviewer Discussion:**

Malware analysis has been a focus of research for decades, and reviewers have questioned the impact of the proposed new dataset. Merely adding new software and non-repetitive samples does not, on its own, establish the dataset's importance. The authors' response was less convincing, as any dataset can become outdated and require updates over time. However, updating data samples alone does not inherently advance a well-studied field unless the scale of the dataset undergoes a significant transformation (e.g., as seen in the ImageNet work) or it provides researchers with novel insights previously unrecognized.

---

### Decision · Program_Chairs · 2025-01-22

Reject